# *Porphyromonas gingivalis* W83 Membrane Components Induce Distinct Profiles of Metabolic Genes in Oral Squamous Carcinoma Cells

**DOI:** 10.3390/ijms23073442

**Published:** 2022-03-22

**Authors:** Sabine Groeger, Jens Martin Herrmann, Trinad Chakraborty, Eugen Domann, Sabine Ruf, Joerg Meyle

**Affiliations:** 1Department of Periodontology, Justus-Liebig-University of Giessen, 35392 Giessen, Germany; jens.m.herrrmann@dentist.med.uni-giessen.de (J.M.H.); joerg.meyle@dentist.med.uni-giessen.de (J.M.); 2Department of Orthodontics, Justus-Liebig-University of Giessen, 35392 Giessen, Germany; sabine.ruf@dentist.med.uni-giessen.de; 3Institute of Medical Microbiology, Justus-Liebig-University of Giessen, 35392 Giessen, Germany; trinad.chakaraborty@mikrobio.med.uni-giessen.de; 4DZIF—Germen Centre for Infection Research, Partner Site Giessen-Marburg-Langen, 35392 Giessen, Germany; eugen.domann@mikrobio.med.uni-giessen.de; 5Institute of Hygiene and Environmental Medicine, Justus-Liebig-University of Giessen, 35392 Giessen, Germany

**Keywords:** *P. gingivalis*, membrane, oral carcinoma cells, transcriptomics, metabolic response, periodontitis, gene expression, cancer

## Abstract

Periodontitis, a chronic inflammatory disease is caused by a bacterial biofilm, affecting all periodontal tissues and structures. This chronic disease seems to be associated with cancer since, in general, inflammation intensifies the risk for carcinoma development and progression. Interactions between periodontal pathogens and the host immune response induce the onset of periodontitis and are responsible for its progression, among them *Porphyromonas gingivalis* (*P. gingivalis*), a Gram-negative anaerobic rod, capable of expressing a variety of virulence factors that is considered a keystone pathogen in periodontal biofilms. The aim of this study was to investigate the genome-wide impact of *P. gingivalis* W83 membranes on RNA expression of oral squamous carcinoma cells by transcriptome analysis. Human squamous cell carcinoma cells (SCC-25) were infected for 4 and 24 h with extracts from *P. gingivalis* W83 membrane, harvested, and RNA was extracted. RNA sequencing was performed, and differential gene expression and enrichment were analyzed using GO, KEGG, and REACTOME. The results of transcriptome analysis were validated using quantitative real-time PCR with selected genes. Differential gene expression analysis resulted in the upregulation of 15 genes and downregulation of 1 gene after 4 h. After 24 h, 61 genes were upregulated and 278 downregulated. GO, KEGG, and REACTONE enrichment analysis revealed a strong metabolic transcriptomic response signature, demonstrating altered gene expressions after 4 h and 24 h that mainly belong to cell metabolic pathways and replication. Real-time PCR of selected genes belonging to immune response, signaling, and metabolism revealed upregulated expression of CCL20, CXCL8, NFkBIA, TNFAIP3, TRAF5, CYP1A1, and NOD2. This work sheds light on the RNA transcriptome of human oral squamous carcinoma cells following stimulation with *P. gingivalis* membranes and identifies a strong metabolic gene expression response to this periodontal pathogen. The data provide a base for future studies of molecular and cellular interactions between *P. gingivalis* and oral epithelium to elucidate the basic mechanisms of periodontitis and the development of cancer.

## 1. Introduction

Periodontitis is an inflammatory disease that affects the periodontal tissues leading to attachment loss, destruction of the periodontal ligament, and the resorption of the alveolar bone [1,2].

The main factor that provokes local inflammation is the development and persistence of a well-organized, various species-harboring microbial biofilm on the surface of the teeth, within the gingival crevice, and on the adjacent epithelium.

As part of the immune response, pro-inflammatory cytokines and lysosomal enzymes are released by various cells, such as neutrophils, macrophages, B-/plasma-cells, keratinocytes, fibroblast, and osteoclasts [3].

In this environment, fibroblasts from the subepithelial connective tissue produce enzymes, mostly metalloproteinases. Together with a concomitant decrease in enzyme inhibitors, an imbalance occurs that initiates destructive processes in the periodontal tissues [4].

It is known that chronic infections are linked to carcinoma development and progression.

Human oral cancers are the world’s 11th most common human neoplasms, with 3% of all newly diagnosed cancer cases [5,6]. More than 90% of them are oral squamous cell carcinomas (OSCC) [7]. Chronic periodontitis is associated with an increased risk for tongue and head and neck carcinomas [8,9].

Among the multiple microorganisms that have been suggested to be involved in the pathogenic process, the Gram-negative anaerobic rod *Porphyromonas gingivalis* (*P. gingivalis*) is considered a keystone pathogen [10]. Some strains express a multitude of virulence factors, including proteins of the outer membrane (OMP), a capsule, proteases such as gingipains, lipopolysaccharides, collagenases, hemolysin, fimbriae, trypsin proteases, and hemagglutinins [11,12,13]. The host’s innate and adaptive immunological reactions can be impaired as well as hyper-reactive. Both types of response result in long-lasting periodontal tissue damage leading to deteriorating situations initiated by the organisms [14]. Innate immune response is primarily activated to counteract antigenic microbial challenges. This activation is required for the subsequent initiation of adaptive immunity. Pathogens exhibit characteristic molecular patterns that are recognized by a variety of cell-surface or intracellular receptors, pattern recognition receptors (PRRs), among them toll-like receptors (TLRs), nucleotide-binding oligomerization domain-containing (NOD)-like receptors (NLRs), C-type lectin receptors (CLRs), and retinoic acid-inducible gene (RIG)-I-like receptors (RLRs). Binding of the pathogen-associated molecular patterns (PAMPs) to these receptors activates intracellular signaling pathways. The activation subsequently stimulates the expression of a number of inducible co-stimulators and induces the release of chemotactic and inflammatory molecules [15].

A number of chemokines participate in the immune response, including the CC chemokine receptor–ligand pair, CC chemokine receptor 6 (CCR6), and CC chemokine ligand 20 (CCL20) cytokines that are prominent targets in immunological research because of their therapeutic potential. CCL20 (MIP3a), a small molecule belonging to the CC chemokine family, shows strong chemotactic properties with respect to lymphocytes, weakly attracts neutrophils, and is a chemokine working on dendritic cells that can recruit Th17 as well as Treg cells to sites of inflammation [16]. CCR6, the binding partner of CCL20, is a member of the G protein-coupled receptor superfamily that is expressed on various T-cells (Th1, Th2, Th17, Treg), B cells, immature dendritic cells (DC), natural killer cells (NKT cells), and also on neutrophils [17]. CCR6 and CCL20 are involved in the coordination of immune homeostasis as well as immune activation. The immunological importance of these chemokine-receptor–ligand interactions is well established since they affect human health and disease. Their effects have important consequences that can appear in multiple organs of the body. Various studies demonstrated that the CCR6 and CCL20 axis directly influences the excretory, respiratory, gastrointestinal, reproductive, nervous, and skeletal systems. Therefore, the CCL20-CCR6 receptor–ligand pair provides a promising therapeutic target. Inhibition or blockade of either of these molecules carries a high potential for successful pharmacological treatment of related diseases [18,19,20].

The liver is one of the primary sites of drug metabolism, influencing oral bioavailability and clearance. The main group of drug-metabolizing enzymes is the cytochrome P450 (P450) group. Cytochrome P450, family 1, subfamily A, polypeptide 1 (CYP1A1) is a member of the cytochrome P450 (CYP P450) family of enzymes. It belongs to a group of mixed-functions mono-oxygenases that are responsible for the phase I oxidative metabolism of a great number of structurally diverse substrates. CYP P450 proteins catalyze several important endogenous biochemical processes in mammalian cells, such as steroid hormone, prostaglandin, and leukotriene biosynthesis [21]. Metabolic processes, including the metabolism of both endobiotics and xenobiotics, are maintained by the liver; the CYP P450 family of proteins participate in these reactions, mostly resulting in mainly non-toxic quantities of metabolites with expedient manners of excretion. Only a subset of these processes results in toxic components, possibly causing direct liver damage. Inflammation throughout liver injury is able to affect the expression as well as the activity of CYPs, possibly by modulating the drug progressing metabolisms. In addition, CYP 450 directly impacts toxicity more than impairment or enhancement of metabolism. CYP-mediated activation of numerous different compounds may affect immune cells, but vice versa, persistent inflammation accompanied by cytokine production induces altered expression levels of CYPs. The numerous interactions between inflammation and CYP activity in the liver are reviewed in Woolbright et al., 2015 [22].

The aims of this study were to investigate the genome-wide impact of *P. gingivalis* W83 membrane on the RNA expression of oral squamous carcinoma cells by transcriptome analysis and to validate the expression of chosen genes by quantitative real-time PCR.

## 2. Results

Analysis of the differential expression after 4 h stimulation with total bacterial membranes revealed that the expression of 15 genes was upregulated and of 1 gene downregulated as compared with negative control (Figure 1A). After 24 h of stimulation 61 genes were upregulated and 278 genes downregulated Figure 1B.

Co-expression analysis revealed that after 4 h, 269 genes were co-expressed from non-stimulated as well as stimulated cells, 346 genes were expressed by non-stimulated cells only, and 364 by TM infected cells only (Figure 2A). The 24 h stimulation showed 147 co-expressed genes, 368 were expressed by non-stimulated cells only, and 248 by TM infected cells only (Figure 2B).

Genes that were associated with distinct biological processes were clustered. GO analysis (Figure 3) revealed that after 4 h (Figure 3A), genes were significantly enriched that are involved in biological processes such as oxidoreductase activity (n = 4), secondary metabolic processes (n = 2), and involvement in the formation of cellular structures (n = 5).

After 24 h (Figure 3B), a significant enrichment of genes could be demonstrated, which are associated with responses to the nucleolus (n = 49), RNA binding (n = 54), tRNA metabolic process (n = 15), anatomical structure formation involvement (n = 38), cell adhesion (n = 42), cellular amino acid metabolic process (n = 17), cell proliferation (n = 49), and cell death (n = 52).

KEGG analysis (Figure 4) revealed that after 4 h of stimulation with TM (Figure 4A), genes were enriched that are related to the following biological functions: chemical carcinogenesis (n = 4), metabolism of xenobiotics by cytochrome P (n = 4), phenylalanine metabolism (n = 2), histidine metabolism (n = 2), β-alanine metabolism (n = 2), tyrosine metabolism (n =2), tryptophan metabolism (n = 2), ovarian steroid genesis (n = 2), steroid hormone biosynthesis (n = 2), drug metabolism (cytochrome P450) (n = 2), glycolysis/gluconeogenesis (n = 2), glycosaminoglycan biosynthesis (heparan) (n = 1), metabolic pathways (n = 3), hedgehog signaling pathway (n = 1), basal cell carcinoma (n = 1), and retinol metabolism (n = 1). After 24 h (Figure 4B) genes were enriched that belong to ribosome biogenesis in eukaryotes (n = 7), focal adhesion (n = 10), proteoglycans in cancer (n = 8), MAPK signaling pathway (n = 9), metabolic pathways (n = 23), leukocyte transendothelial migration (n = 6), regulation of actin cytoskeleton (n = 8), steroid synthesis (n = 3), steroid hormone biosynthesis (n = 4), aminoacyl-tRNA biosynthesis (n = 4), arhythmogenic right ventricular cardiomyopathy (n = 4), alanine, aspartate and glutamate metabolism (n = 3), and RNA degradation (n = 4).

The analysis using REACTOME (Figure 5) showed that after 4 h of incubation genes related to the following biological processes were enriched (Figure 5A): phase I functionalization of compounds (n = 4), biological oxidation (n = 4), cytochrome P450 (n = 2), arachidonic acid metabolism (n = 2), regulation of lipid metabolism by peroxisomes (n = 2) and PPARA activated gene expression (n = 2). After 24 h (Figure 5B), genes for rRNA modification in the nucleus and cytosol (n = 12), rRNA processing (n = 17), rRNA processing in the nucleus and cytosol (n = 16), major pathway of rRNA processing in the nucleus (n = 15), tRNA modification in the nucleus and cytosol (n = 7), formation of cornified envelope (n = 9), RAF-independent MAPK1/3 activation (n = 4), and tRNA processing (n = 8) were enriched.

The results from qPCR (Figure 6) show that differences in the chosen genes occurred after stimulation with TM in SCC-25 cells (Figure 6A). After 4 h, a 6.8-fold upregulation of CCL20, 6-fold of chemokine C-X-C motif (CXCL)8, 4.5-fold of NF-κB inhibitor alpha (NFκBIA), 3.8-fold of TNFAIP3, 1.5-fold of TRAF5, 3.7-fold of CYP1A1, and 1.7-fold of NOD2 were detected. After 24 h (Figure 6B), the following cytokines were upregulated: CCL20 1.6-fold, CXCL8 3.7-fold, NFκBIA 2.2-fold, TNFAIP3 2.9-fold, TRAF5 1.9-fold, CYP1A1 3.3-fold, and NOD2-1.5 fold upregulated. So, all the chosen genes were shown to be upregulated, and the results confirmed the data from the transcriptomic analysis.

## 3. Discussion

Complete expression profiles of oral epithelial carcinoma cells in response to stimulation with *P. gingivalis* membranes have not been described so far. In this study, RNA-seq was used to assess the complete transcript that is induced after stimulation with *P. gingivalis* W83 total membrane extracts (TM).

RNA-seq is accounted for being a powerful digital instrument for analysis of gene expression and is capable of sensitive, unbiased, and comprehensive transcriptomic analysis [23]. This study presents a transcriptomic snap-shot of the epithelial cell reaction to infection with bacterial TM. RNA sequencing of total RNA extracted from SCC-25 cell isolates is a valuable tool for further differential expression, gene ontology, pathway, and network enrichment analyses. The results revealed molecular responses in the total RNA transcriptome, demonstrating a significantly differential expression of a number of genes. Gene ontology and pathway studies of differentially expressed total RNA identified not only, as expected from previous studies [24,25,26,27], an immune responsive signature but also cell metabolic activity. Transcripts in SCC-25 cells of distinct genes were validated by RT-qPCR.

The human oral squamous cell carcinoma cell total RNA transcriptome was clearly impacted by infection with *P. gingivalis* TM.

Stimulation with TM for 4 h appeared to drive the epithelial cell toward a molecular state that might promote enzyme activity but also inflammation. Gene ontology and pathway analysis of the RNA-seq data revealed activity in the mitochondrial segment and immune response signaling-related categories, and some genes identified in the network analysis were transcriptional regulators linked to metabolic processes and inflammation, including members of the NF-kB family and oxidoreductases such as cytochrome C.

CCL20 and CXCL8 both function as strong immune-modulating molecules. Their expression is altered by *P. gingivalis* peptidylarginine deiminase [28]. It was demonstrated that elevated CCL20 expression was induced in epithelial cells after infection with oral bacteria [29,30]. Furthermore, in periodontitis tissue samples, CCL20 levels were shown to be elevated [31].

Some of the genes affected by TM were related to regulation and control of the nuclear factor kappa B (NF-κB) pathway, a major activator of genes in both innate and acquired immune responses through binding to an interferon-stimulated response element (ISRE). ISRE promoters include the tumor necrosis factor-inducible gene (TNFAIP)3 [32]. TNFAIP3 was not demonstrated to be related to oral epithelial cell biology. However, the results of a few studies suggest that it may have a suppressive effect on osteoclastogenesis [33]. Enhanced TNFAIP3 in gingival tissue was demonstrated to be linked to reduced periodontal inflammation and TLR9 activity [34].

NF-κB is the main transcription regulator of cell adhesion, proliferation, immune response, apoptosis, and differentiation [35]. In non-activated cells, NF-κB is inactive in the cytoplasm associated with a sequestering inhibitory protein, IκBα, β or γ. The most current protein of this family of inhibitory proteins is the NF-κB inhibitor α (NF-κBIA) [36].

It has been reported that NF-κB1 and NF-κBIA polymorphisms probably conjointly contribute to the risk of colorectal cancer [37]. Furthermore, it was demonstrated that the NF-κB1 and NF-κBIA polymorphisms may have a role in lung carcinogenesis and prognosis [38].

CXCL8 belongs to the chemotactic cytokine (chemokine) family of proteins, which are important in regulating cell migration and play a role in mediating leukocyte cell recruitment by chemokine receptor signaling on the cell surfaces [39,40,41].

TSG-6 was demonstrated to inhibit chemokine-induced transendothelial migration of neutrophils via direct interaction between the glycosaminoglycan binding site of CXCL8 and TSG-6. TSG-6 was also found to affect the binding of CXCL8 to cell surface glycosaminoglycans and the subsequent transport of CXCL8 across an endothelial cell monolayer, so TSG-6 thereby was identified as a CXCL8-binding protein [42].

Tumor necrosis factor receptor (TNFR)-associated factors (TRAFs), including TRAF1, 2, 3, 5, and 6, are molecular activators for TNFRs signaling pathways and interleukin-1 receptor-associated nuclear factor kappa-light-chain-enhancer of activated B cells (NF-κB) signaling pathway, acting as inducers of cell death or necrosis [43,44,45].

Among them, TRAF5 plays a role in the regulation of the Notch signaling pathway-related NF-κB activation in brain cancer [46]. Micro RNA miR-26b was downregulated through inverse regulation of TRAF5 as its downstream target. This process was shown to provide tumor-suppressive effects on human esophageal squamous carcinoma cells (ESCC) over the induction of inhibited cellular proliferation, cell-cycle transition, and migration [47]. Shang et al. (2012) demonstrated that HP1BP3, a ubiquitously expressed nuclear protein belonging to the H1 histone family, significantly increased tumor metastasis and growth in ESCC cells. The authors reported that HP1BP3 regulates these functions by upregulating the micro RNA miR-23a, which targets TRAF5 downstream to affect cell survival and proliferation [48].

With regards to immunometabolism, two main sectors of the field have risen: namely, how obesity influences the immune system and promotes inflammation, triggering pancreatic islet failure and/or insulin resistance (IR). The second sector addresses the query of which intrinsic metabolic changes occur in the immune cell that expedite immune system activation [49].

A genome-wide association study addressed candidate genes, functional elements, and pathways in patients with different severities of chronic periodontitis (CP) using gene ontology, ingenuity, KEGG, Panther, Reactome, and Biocarta databases for gene set enrichment analysis. It is obvious that the top gene sets were endoplasmic reticulum membrane, cytochrome P450, microsome, and oxidation-reduction gene expression [50]. In concordance with our study, these functional units include genes belonging to cell metabolic processes. The object of another study was to investigate the roles of genetic polymorphisms of metabolizing enzymes as risk factors for periodontitis, including CYP1A1. This study group demonstrated a significantly increased risk for periodontitis in individuals exhibiting the polymorphic CYP1A1 m2 allele [51].

The processes of immune response are closely entangled with metabolic processes.

Uric acid is a metabolite that may act as a danger-associated molecular pattern (DAMP). It is able to induce inflammatory responses under sterile conditions.

It was shown that *P. gingivalis* gingipains have an impact on the THP-1 macrophage uric acid production by enhancing the expression and activity of xanthine oxidoreductase (XOR). Uric acid sodium salt induces cell death, caspase-1 activation, and the expression of pro-inflammatory cytokines, including IL-1α, IL-6, and IL-8, in a human keratinocyte (HOK-16B) cell line. These results suggest that gingipain-induced uric acid possibly mediates inflammation in cells of periodontal tissues [52].

In allergic lung inflammation, the expression of IL-4, IL-13, and CCL11 increased during the acute phase of allergic inflammation and decreased with its resolution. The expression of CCL20 was enhanced during the resolution phase. A differential expression pattern was shown for the CYP gene family that correlated with the state of the inflammatory response [53]. In patients with rheumatoid arthritis, a negative correlation of the levels of IL-1RA, IL-6, and CXCL8 with the expression of the CYP3A4 phenotype of the CYP family was detected [54]. Sen et al., 2020, investigated the effect of global gene expression of HPV-positive and negative cervical cancer cells after stimulation with miR-214 micro RNA via next-generation sequencing and validated the results with 11 selected relevant genes, including TNFAIP3 and CYP1B1, using quantitative real-time PCR [55].

Xiao et al., 2014, evaluated the therapeutic effects of zeaxanthin dipalmitate (ZD) on a rat alcoholic fatty liver disease (AFLD) model. Treatment with ZD induced a lower expression level of cytochrome P450 2E1 (CYP2E1) and diminished the activity of nuclear factor kappa B (NF-κB) through the restoration of its inhibitor kappa B alpha (IκBα) [56].

Genetic variations may affect inflammatory responses depending on their influence on protein activity and the relevance of such proteins in the pathway of the drug. Boso et al., investigated a panel of 33 single nucleotide polymorphisms in 14 different genes encoding for the most relevant metabolizing enzymes and drug transporters. The results revealed a number of allele frequencies. The largest differences occurred in six genes, including CYP3A5 and NOD2 [57]. All these studies demonstrate the strong linkage between immunologic and metabolic processes and cell responses.

In summary, our results provide insight into the RNA transcriptome of human oral squamous carcinoma epithelial cells upon infection with *P. gingivalis* TM and discovered distinct immunological and metabolic gene expression responses to these molecules.

In conclusion, the data serve as a base for future studies of molecular and cellular interactions between *P. gingivalis* and oral carcinoma cells to shed light on basic mechanisms in the development of carcinomas and especially in the presence of periodontitis.

As with the majority of studies, there are some limitations in this study. The main limitation is the usage of only one carcinoma cell line for the transcriptomic analysis. Since transcriptomics is a rather complex method that requires time and resources, the usage of just one cell type is not unusual. Future research will address the investigation of different cell types.

## 4. Materials and Methods

### 4.1. Preparation of P. Gingivalis Total Membrane

Bacteria were harvested in the late exponential growth phase (OD600 of 1.0) by centrifugation for 20 min at 6500× *g* at 25 °C. The pellet was resuspended in 50 mL of 10 mM HEPES, pH 7.4, containing a protease inhibitor cocktail (4 mini-tablets of complete, EDTA-free, Roche) and DNase I/RNase A (20 μg/mL each).

Bacteria were disrupted by four passages through a high-pressure cell disruption system (Model TS, 0.75 KW, Constant Systems Ltd., Daventry, UK) at 40,000 psi. The cellular debris was removed by centrifugation at 8000× *g* for 30 min at 4 °C. The membranes were sedimented from the cleared lysate at 150,000× *g* for 2 h at 4 °C. The supernatant (cytosolic fraction) was stored, and the total membrane fraction was washed three times with 10 mM HEPES, pH 7.4. The membrane pellet was subsequently resuspended in 10 mM HEPES, pH 7.4.

### 4.2. Cell Culture

The human oral tongue squamous cell carcinoma cell line SCC-25 was purchased from the DSMZ (German Collection of Microorganisms and Cell Cultures, Braunschweig, Germany, DSMZ numbers ACC 617). Cells were cultured in a medium containing Dulbecco’s minimal essential medium (DMEM): Ham’s F12 (4:1 *v*/*v*), Hepes buffer, (Invitrogen, Karlsruhe, Germany) and 10% fetal calf serum (FCS, Greiner, Frickenhausen, Germany). The cells were seeded in 6-well plates at 1 × 10^6^ cells per well and grown at 37 °C in a humidified atmosphere with 5% CO_2_ to 80% confluency before stimulation. The cells were stimulated using 100 µg/mL TM for 4 and 24 h.

### 4.3. RNA Extraction

After the incubation, the cells were lysed, and the RNA was extracted using NucleoSpin^®^ RNA Plus (Machery-Nagel, Munich, Germany) columns and solutions following the manufacturer’s instructions. The lysis was performed using the included lysing buffer that contains guanidinium thiocyanate (Machery-Nagel, Düren, Germany). The eluted RNA purity and quantity of each sample were verified photometrically by optical density (OD) readings of the A260/280 nm ratio (Nanodrop 2000, Thermo Fisher Scientific, Waltham, MA, USA). The spectrophotometric ratio of A260/280 varied >1.80, and A260/230 values yield a ratio >2.0, so the isolated RNA could be regarded as pure.

### 4.4. RNA-Seq

The RNA-seq was performed by NOVOGEN (Cambridge, UK). The process was the following: After quality control of the RNA, mRNA from eukaryotic organisms was enriched using oligo (dT), fragmented randomly in fragmentation buffer, followed by cDNA synthesis using random hexamers and reverse transcriptase. After first-strand synthesis, the second strand was synthesized by nick-translation. The final cDNA library is ready after a round of purification, terminal repair, A-tailing, ligation of sequencing adapters, size selection, and PCR enrichment. Library concentration was first quantified, and then sequencing was performed using an Illumina device. After obtaining the results, analysis of differential expression analysis was performed. Cluster analysis was performed to find genes with similar expression patterns under various experimental conditions. Enrichment analysis of the differential expressed genes was performed to find out which biological functions or pathways are significantly associated with differentially expressed genes. Gene Ontology (GO, http://geneontology.org/docs/ontology-documentation/ accessed on 16 March 2022) enrichment analysis was the next step. GO enrichment analysis is used by GOseq, which is based on Wallenius non-central hyper-geometric distribution. Kyoto Encyclopedia of Genes and Genomes (KEGG) enrichment analysis was performed to find out interactions of multiple genes that may be involved in certain biological functions such as enriched metabolic pathways or signal transduction pathways associated with differentially expressed genes compared with the whole genome background. A further instrument for the interpretation of the results is analysis using REACTOME, a peer-reviewed bioinformatics database for biological processes and pathways.

### 4.5. Quantitative Real-Time PCR

Expression of mRNA was assayed 4 and 24 h after infection. The cDNA synthesis was performed with the Verso™ cDNA Kit (Thermo Fisher Scientific, Dreieich, Germany). Quantitative real-time polymerase chain reaction (RT-qPCR) was performed with the SensiFast no ROX SYBR Green Mix according to the manufacturer’s instructions (Bioline, Luckenwalde, Germany). The following primers were used: QuantiTect Primer Assay (Qiagen, Hilden, Germany) Hs_CCL20_1_SG (CCL20), Hs_CXCL8_1_SG (CXCL8), Hs_NFκBIA_1_SG (NFκBIA), Hs_TNFAIP3_1_SG (TNFAIP3), Hs_TRAF5_1_SG, Hs_CYP1A1_1_SG, and Hs_NOD2_1_SG as target genes and Hs_GAPDH_1-SG (GAPDH) as housekeeping gene (patents: Roche Molecular Systems, Pleasanton, CA, USA). The cycling conditions were the following: 1. 95 °C for 2 min; 2. 95 °C for 5 s; 3. 60 °C for 10 s; 4. 72 °C for 20 s, plate read; 5. 39 more cycles from 2; 6. melt curve 60° to 95 °C, increment 0.05° for 5 s, then final plate read.

Cycling and detection was performed in a Bio-Rad CX96 cycler (Bio-Rad, Feldkirchen, Germany). The values were analyzed using the comparative CT (ΔΔCT) method. The amount of target (2^−ΔΔCT^) was obtained normalized to GAPDH and relative to infected cells.

## Figures and Tables

**Figure 1 ijms-23-03442-f001:**
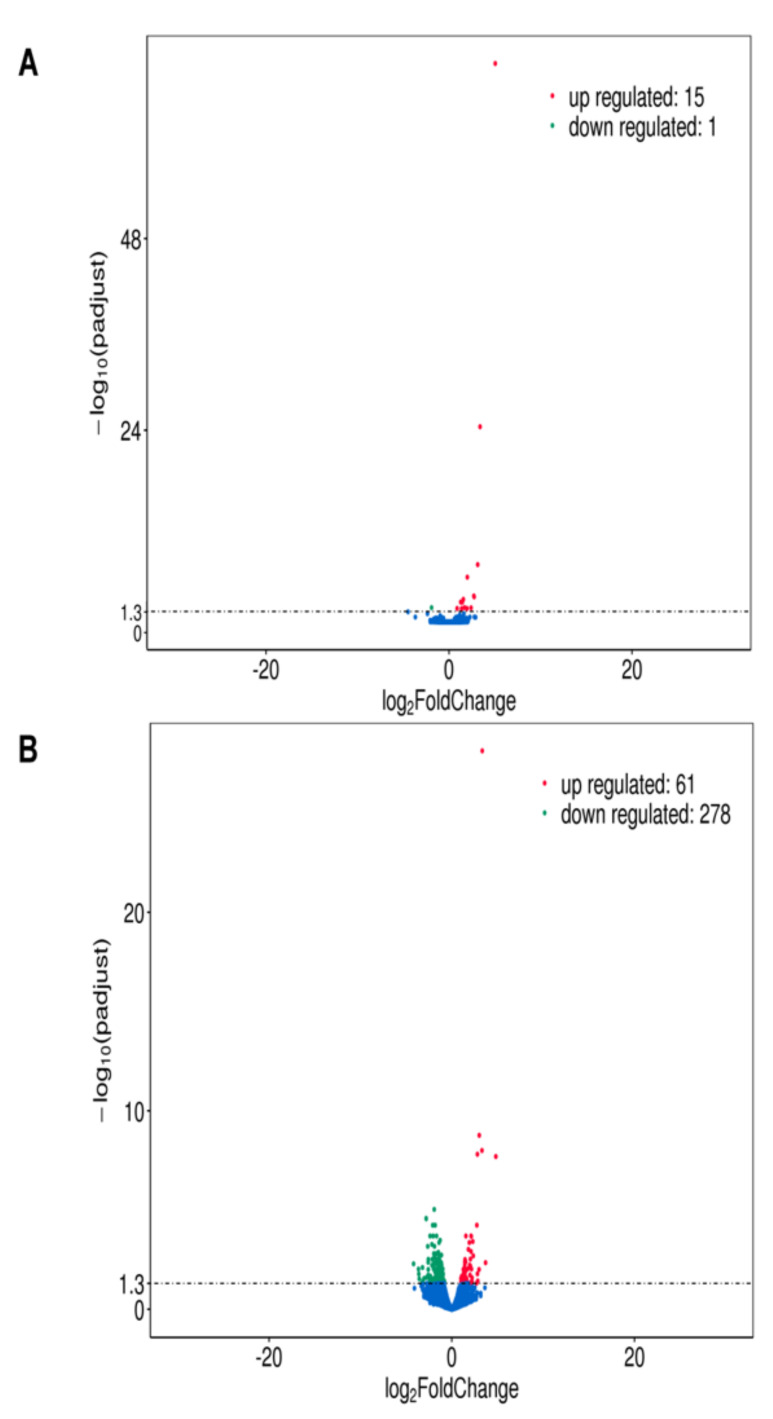
Differential expression analysis of TM stimulated SCC-25 cells compared to the non-stimulated negative control. The input data for differential gene expression analysis are read counts from gene expression level analysis. The differential gene expression analysis contains read counts normalization, model-dependent *p*-value estimation, and FDR value estimation based on multiple hypothesis testing. The results are shown as volcano plots; the changes are indicated as log2fold change, padjust is the normalized *p*-value. NC = non-stimulated cells = negative control, TM = TM stimulated cells: (**A**) stimulation for 4 h, red dots: upregulated genes (15), blue dots: downregulated genes (1); (**B**) stimulation for 24 h red dots: upregulated genes (61), blue dots: downregulated genes (278).

**Figure 2 ijms-23-03442-f002:**
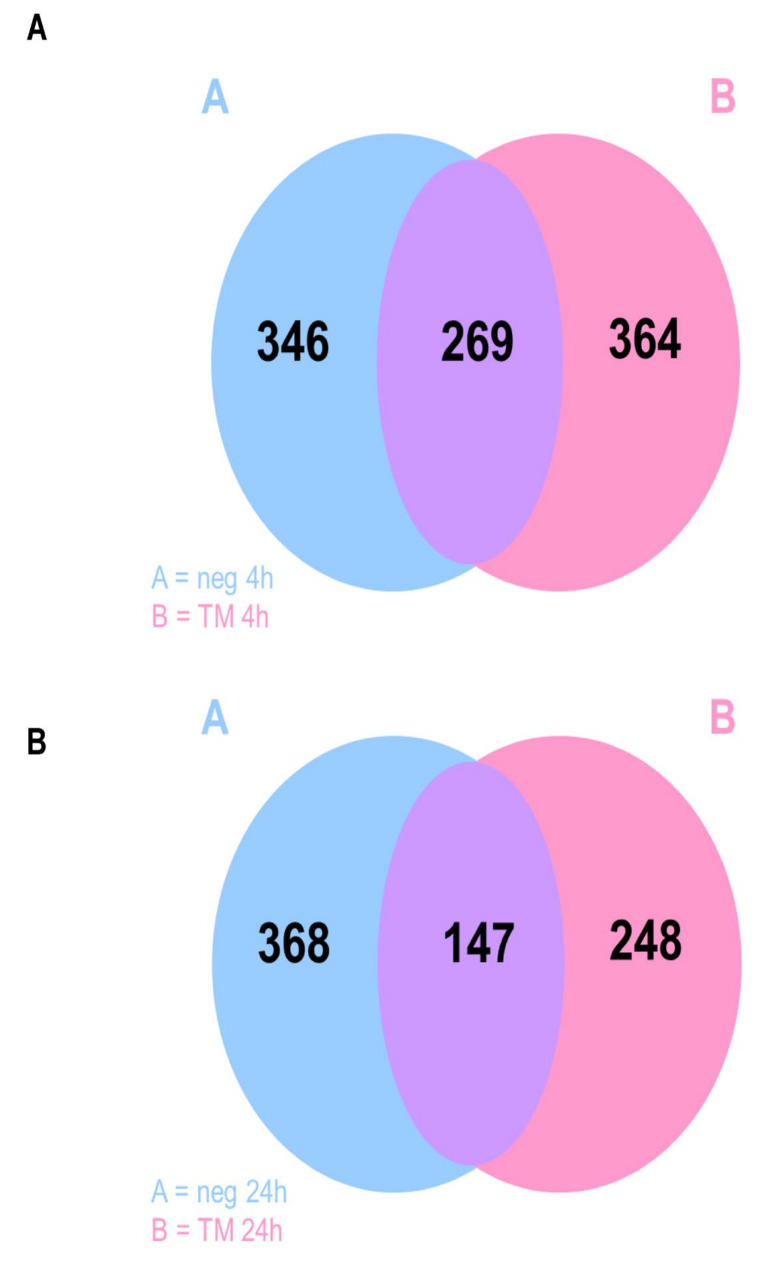
Co-expression of genes expressed by SCC-25 cells are shown as Venn diagrams; neg = non-stimulated cells, negative control, TM = TM stimulated cells: (**A**) stimulation for 4 h; (**B**) stimulation for 24 h.

**Figure 3 ijms-23-03442-f003:**
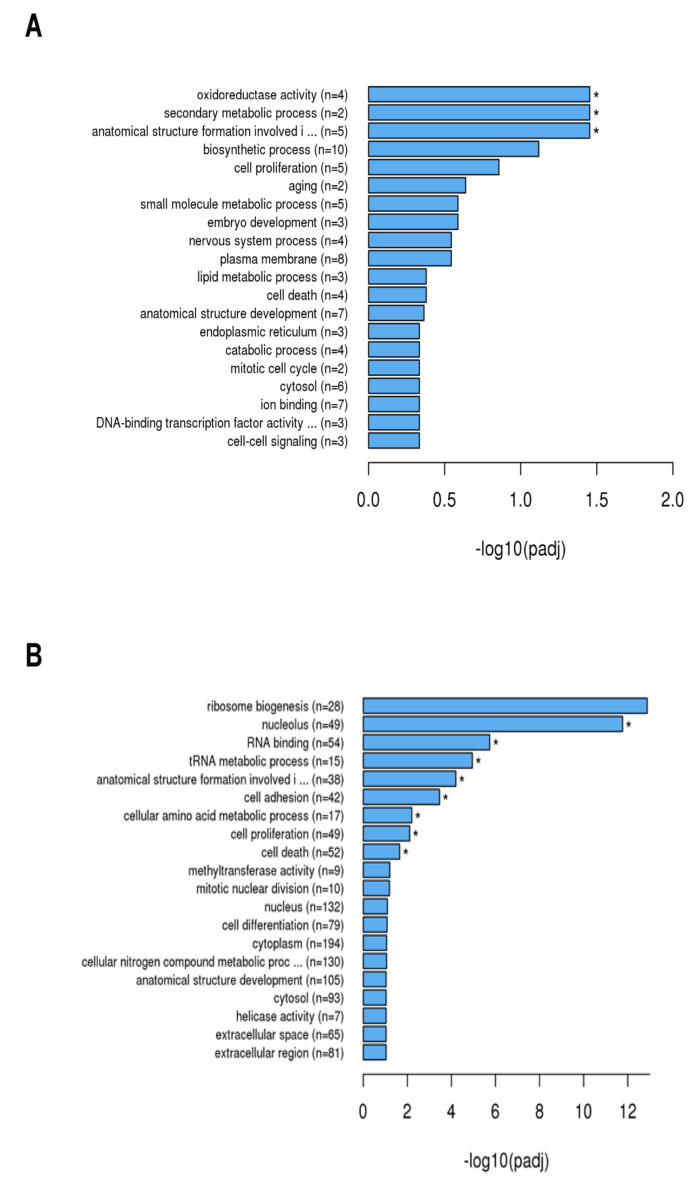
Enrichment analysis using Gene Ontology (GO) of the differential expressed genes by SCC-25 cells was performed to find out which biological functions or pathways are significantly associated with differentially expressed genes. The noted gene expressions of biological processes, molecular functions, and cellular components are exhibited in a directed acyclic graph structure: (**A**) = 4 h stimulation with TM; (**B**) = 24 h stimulation with TM; padj = adjusted *p*-value. Generally, GO terms with corrected *p*-value shown as * < 0.05 (significant enrichment).

**Figure 4 ijms-23-03442-f004:**
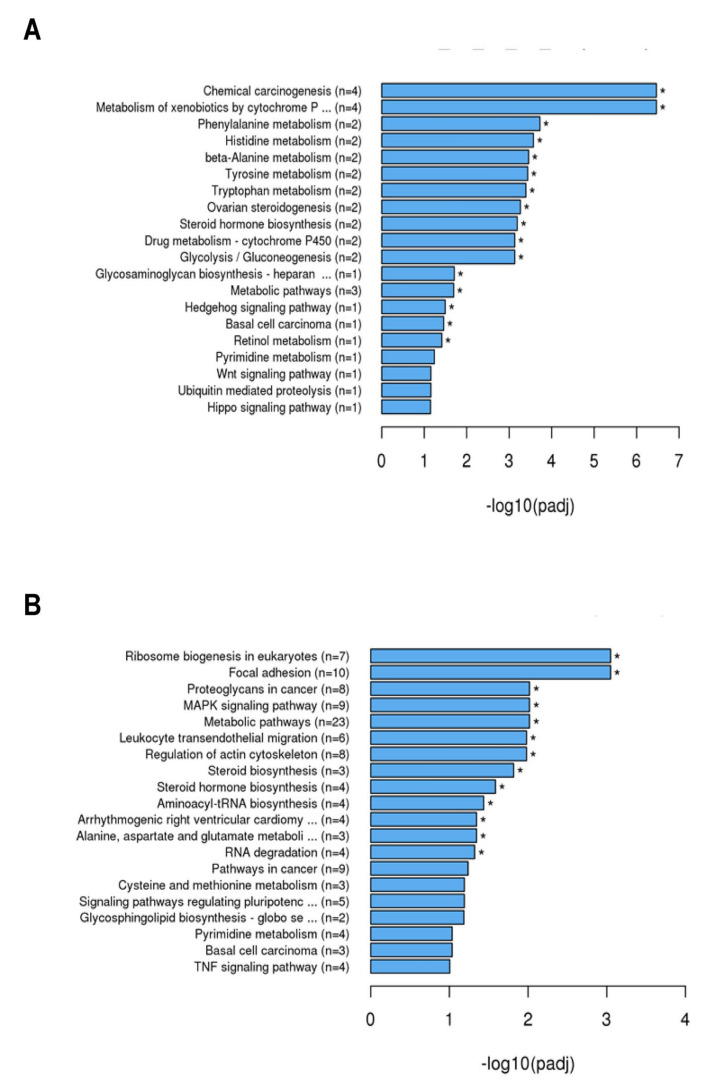
Kyoto Encyclopedia of Genes and Genomes (KEGG) analysis of the differential expressed genes by SCC-25 cells was performed to find out interactions of multiple genes that may be involved in certain biological functions such as enriched metabolic pathways or signal transduction pathways associated with differentially expressed genes. The explored gene expressions of pathways and signal transduction are depicted in the graph: (**A**) 4 h stimulation with TM; (**B**) 24 h stimulation with TM; padj = adjusted *p*-value. Generally, GO terms with corrected *p*-value shown as * < 0.05 (significant enrichment).

**Figure 5 ijms-23-03442-f005:**
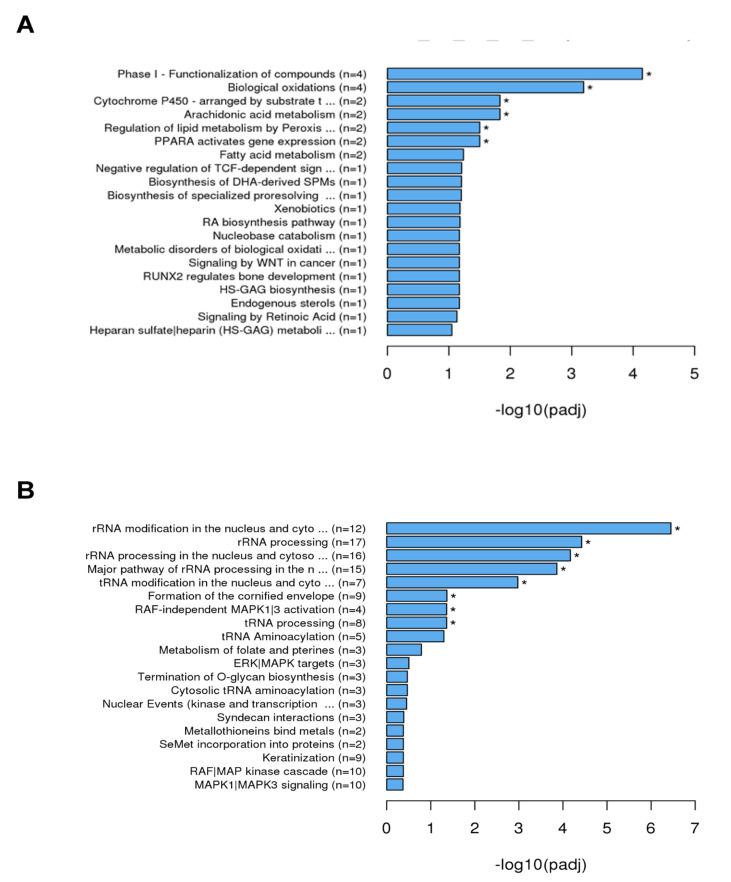
REACTOME analysis of the differential expressed genes by SCC-25 cells was performed to find out interactions of multiple genes that may be involved in certain biological functions such as enriched metabolic pathways or signal transduction pathways associated with differentially expressed genes REACTOME annotates genes to peer-reviewed pathways. The explored gene expressions of pathways and signal transduction are illustrated in graph structure: (**A**) 4 h stimulation with TM; (**B**) 24 h stimulation with TM; padj = adjusted *p*-value. Generally, GO terms with corrected *p*-value shown as * < 0.05 (significant enrichment).

**Figure 6 ijms-23-03442-f006:**
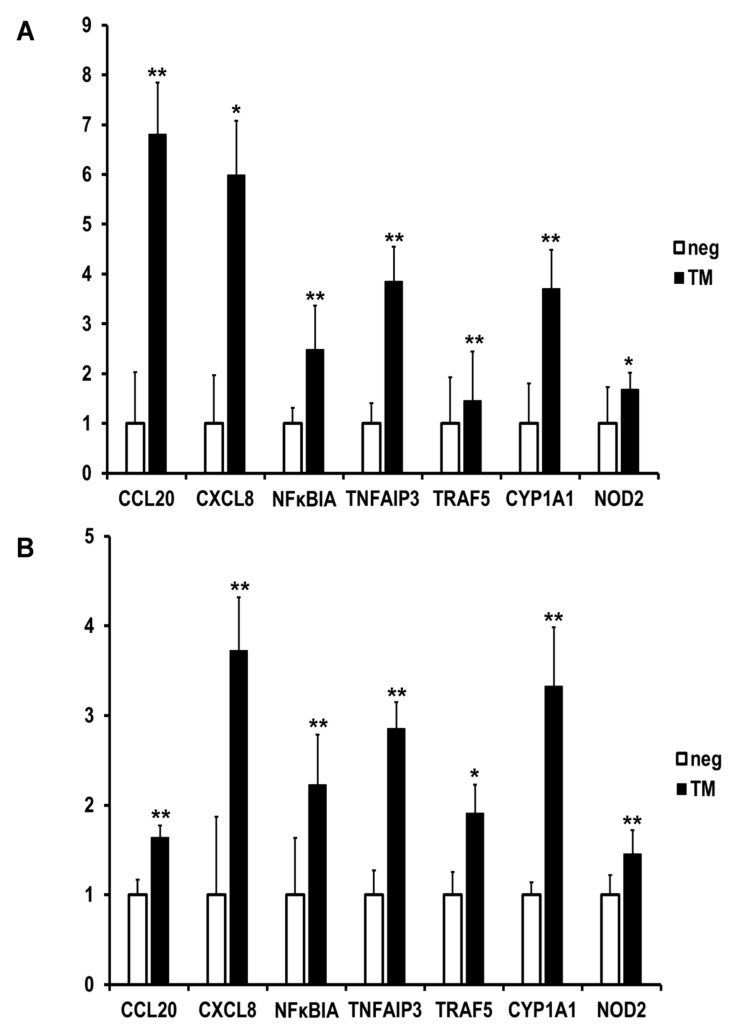
Validation of the results of the RNA-seq using quantitative real-time PCR with 5 chosen genes analyzed by the ΔΔCt method, shown as absolute fold induction relative to infected, normalized to GAPDH, * *p* < 0.05, ** *p* < 0.01 (n = 3), A = gene expression after 4 h, B = gene expression after 24 h. Expressions of the following genes were analyzed: CCL20 = chemokine (C-C motif) ligand 20 (CCL20), CXCL8 = chemokine (C-X-C motif) ligand 8, NFKBIA = nuclear factor kappa B inhibitor alpha, TNFAIP3 = tumor necrosis factor alpha-inducible gene 3 protein, TRAF5 = TNF receptor-associated factor 5, CYP1A1 = cytochrome P450 family 1 subfamily A member 1, NOD2 = nucleotide-binding oligomerization domain-containing 2.

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
