# Peer review of "Porphyromonas gingivalis W83 Membrane Components Induce Distinct Profiles of Metabolic Genes in Oral Squamous Carcinoma Cells"

_ijms, 2022, doi:10.3390/ijms23073442_

Round 1

Reviewer 1 Report

The manuscript “Porphyromonas gingivalis W83 membrane components induce distinct profiles of metabolic genes in oral squamous carcinoma cells” is a research article that investigated the genome-wide impact of Porphyromonas gingivalis W83 membranes on RNA expression of oral squamous carcinoma cells by transcriptome analysis attempted to validate the expression of chosen genes by qRT-PCR. The experimental approach includes only one cell line and a too simplistic methodology.

The manuscript requires an extensive revision. Several grammar mistakes and typos are present, as well as a general construction of sentences that makes the reading not smoothly. Too many very short sentences are present.

There are some concerns which should be addressed:

Major

  1. The abstract is written in with the use of too many concise sentences (e.g. Periodontitis, a chronic inflammatory disease is caused by a bacterial biofilm. All periodontal tissues and structures are affected. This chronic disease seems to be associated with cancer); Please thoroughly re-write it so that reading will be more smoothly.
  2. Authors utilized only one cell line (SCC-25) and they should validate their results utilizing at least one other oral squamous cell line of the tongue (experiments requested).
  3. In qRT-pcr, normalizing was performed using “an endogenous reference (GAPDH) relative to non-infected cells”. This is a methodological mistake. The expression of infected cells should be normalized for (GAPDH) relative to infected cells. Please re-analyze data and modify the results according.
  4. Materials and methods RNA extraction: please specify what buffer what used for lysis, and what values were obtained to assess that the quality of the RNA was acceptable.
  5. Materials and methods RNA seq: please specify the manufacturer of reverse transcriptase.
  6. Quantitative Real-time PCR: Cycling details for each gene analyzed must be specified.
  7. Line 54: “More than 90% of them are oral squamous cell carcinomas (OSCC)”; references cited are of years 1995 and 2002 and should be replaced with a more recent data regarding the incidence of OSCC (PMID: 34827592).

Minor

  1. Line 24 and everywhere else in the text: please replace “real time PCR” by “Real-time PCR”

Author Response

Answer: We thank the reviewer a lot for their helpful comments and suggestions. We addressed them all very carefully and hope, the manuscript now is sufficient for publication.

Reviewer 1: Comments and Suggestions for Authors

The manuscript “Porphyromonas gingivalis W83 membrane components induce distinct profiles of metabolic genes in oral squamous carcinoma cells” is a research article that investigated the genome-wide impact of Porphyromonas gingivalis W83 membranes on RNA expression of oral squamous carcinoma cells by transcriptome analysis attempted to validate the expression of chosen genes by qRT-PCR. The experimental approach includes only one cell line and a too simplistic methodology.

The manuscript requires an extensive revision. Several grammar mistakes and typos are present, as well as a general construction of sentences that makes the reading not smoothly. Too many very short sentences are present. There are some concerns which should be addressed:

Major

  1. The abstract is written in with the use of too many concise sentences (e.g. Periodontitis, a chronic inflammatory disease is caused by a bacterial biofilm. All periodontal tissues and structures are affected. This chronic disease seems to be associated with cancer); Please thoroughly re-write it so that reading will be more smoothly.

Answer: The abstract was rewritten addressing this concern and is more smoothly readable now.

  1. Authors utilized only one cell line (SCC-25) and they should validate their results utilizing at least one other oral squamous cell line of the tongue (experiments requested).

Answer: Transcriptomic analysis is a rather complex method that requires time and resources. The usage of just one cell type is not unusual (Arjunan et al., 2016, Islam et al., 2020). The investigation of primary oral cells and further oral squamous carcinoma cells was performed in previous studies (Groeger et al., 2011, 2017a, 2017b, 2020). Transcriptomic analysis of these cell types will be subject of future studies. This is addressed now in an additional section about the main limitation.

  1. In qRT-pcr, normalizing was performed using “an endogenous reference (GAPDH) relative to non-infected cells”. This is a methodological mistake. The expression of infected cells should be normalized for (GAPDH) relative to infected cells. Please re-analyze data and modify the results according.

Answer: We apologize for this mistake. Of course were the results of the qRT-PCR performed normalized for GAPDH and relative to infected cells. This is corrected now in the manuscript.

  1. Materials and methods RNA extraction: please specify what buffer what used for lysis, and what values were obtained to assess that the quality of the RNA was acceptable.

Answer: This information is provided now in the manuscript.

  1. Materials and methods RNA seq: please specify the manufacturer of reverse transcriptase.

Answer: This information is provided now.

  1. Quantitative Real-time PCR: Cycling details for each gene analyzed must be specified.

Answer: This information is provided now.

  1. Line 54: “More than 90% of them are oral squamous cell carcinomas (OSCC)”; references cited are of years 1995 and 2002 and should be replaced with a more recent data regarding the incidence of OSCC (PMID: 34827592).

Answer: The references were replaced with the suggested one as requested.

Minor

  1. Line 24 and everywhere else in the text: please replace “real time PCR” by “Real-time PCR”

Answer: This was performed accordingly.

Reviewer 2 Report

This work sheds light on the RNA transcriptome of human oral squamous carcinoma cells following stimulation with P. gingivalis membranes and 31
identifies a strong metabolic gene expression response to this periodontal pathogen. Although the information provided in this study and the experimental methodology is interesting, the authors could have explained this manuscript more thoroughly and avoided the language issues. Hence, several major points should be addressed before publication:

  1. Introducion; Are there other methods that should be discussed in the introduction as well, how does this approach compare to those?
  2. Please improve the resolution of all the figures, some figures cannot see it clearly.
  3. The conclusion looks fine, and the main limitation also should be discussed as well.
  4. There are some grammatical errors in this manuscript such as continuously forgetting to add ‘a’ or ‘the’ before a specific word which limits the clarity of the author’s writing. Check the language issues.

Author Response

Answer: We thank the reviewer a lot for their helpful comments and suggestions. We addressed them all very carefully and hope, the manuscript now is sufficient for publication.

Reviewer 2: Comments and Suggestions for Authors

This work sheds light on the RNA transcriptome of human oral squamous carcinoma cells following stimulation with P. gingivalis membranes and 31
identifies a strong metabolic gene expression response to this periodontal pathogen. Although the information provided in this study and the experimental methodology is interesting, the authors could have explained this manuscript more thoroughly and avoided the language issues. Hence, several major points should be addressed before publication:

  1. Introduction; Are there other methods that should be discussed in the introduction as well, how does this approach compare to those?

Answer: Transcriptomic analysis is a well established method that allows comprehensive insight into the reaction of cells to a distinct stimulus. It was chosen by the authors of this work because of this attempt. Investigations that compare different methods are undoubtedly a very interesting field. This was not the primary aim of this study though.

  1. Please improve the resolution of all the figures, some figures cannot see it clearly.

Answer: The resolution of all figures was improved. We hope, that it is appropriate now.

  1. The conclusion looks fine, and the main limitation also should be discussed as well.

Answer: The manuscript was completed with a section about the main limitations as requested.

  1. There are some grammatical errors in this manuscript such as continuously forgetting to add ‘a’ or ‘the’ before a specific word which limits the clarity of the author’s writing. Check the language issues.

Answer: The manuscript was checked carefully in this regard and errors were corrected.

Round 2

Reviewer 1 Report

The manuscript deserves to be published in the present form.

Reviewer 2 Report

The authors have addressed in detail the concerns raised by the referees and the manuscript is suitable for publication.